# Does it work? Using a Meta-Impact score to examine global effects in quasi-experimental intervention studies

**Nancy Elizabeth Doyle** [1]*, **Almuth McDowall**[2], **Raymond Randall**[3], **Kate Knight**[4]

**1** School of Psychology, City, University of London, London, United Kingdom, **2** Department of Organizational Psychology, Birkbeck University of London, London, United Kingdom, **3** Sheffield University Management School, Sheffield, United Kingdom, **4** Independent Maths Consultant, United Kingdom

* n.doyle@bbk.ac.uk

## Abstract

The evaluation of applied psychological interventions in the workplace or elsewhere is challenging. Randomisation and matching are difficult to achieve and this often results in substantial heterogeneity within intervention and control groups. As a result, traditional comparison of group means using null hypothesis significance testing may mask effects experienced by some participants. Using longitudinal studies of coaching interventions designed to provide support for dyslexic employees, this study describes and evaluates a different approach using a Meta-Impact score. We offer a conceptual rationale for our method, illustrate how this score is calculated and analysed, and show how it highlights person-specific variations in how participants react and respond to interventions. We argue that Meta-Impact is an incremental supplement to traditional variable-centric group-wise comparisons and can more accurately demonstrate in practice the extent to which an intervention worked. Such methods are needed for applied research, where personalized intervention protocols may require impact analysis for policy, legal and ethical purposes, despite modest sample sizes.

## Introduction

The evaluation of applied well-being and performance interventions in psychology is beset with methodological challenges, particularly regarding measurement of any impact on important outcomes. The influence of contextual, extraneous variables occurring before and during the intervention period cannot be fully controlled or analysed. Participants enter the intervention with different needs, goals and motivations, facing different demands and coming equipped with different resources. Under the strains of traditional null hypothesis significance testing (NHST) and even with more contemporary approaches that balance *p*-values with nuanced reporting of confidence intervals and effect size, participant attrition and heterogeneity risks a Type II error [1]. Further, current available methods in psychology focus on identifying the average, common mechanisms of an intervention to predict success. In practice, active mechanisms and pathways through an intervention are personalized. Qualitative approaches

**Data Availability Statement:** Raw data are included in the appendices.

**Funding:** ND was funded by Genius Within (Private non-profit entity) to undertake a PhD program within which this study was conceived. However

the funders had no role in the study design, data collection and analysis, decision to publish or preparation of the manuscript.

**Competing interests:** The authors have declared that no competing interests exist.

accommodate personalization but do not easily generalize and cannot predict the success of interventions to inform policy and practice. In this paper, we present a 'Meta-Impact score' conceived to remedy this paradox, in which we determine whether an intervention has an overall effect (i.e. to answer the question 'does it work?') whilst simultaneously taking into account the different ways in which it 'works' for a heterogeneous participant group experiencing the same intervention in different ways. We situate this method in a pragmatic, realist paradigm and propose its deployment to improve primary studies in field settings, in order to deliver ecologically valid psychological advice [2, 3]. We first outline the conceptual problems with existing methods, before providing a detailed, step-by-step account of a field study in which the logic of Meta-Impact was developed.

## Variable-centered, person-centered and person-specific methods

The testing of intervention effects using quantitative evaluation methods has usually relied upon stability and homogeneity within groups and across intervention protocols. Evaluation is often variable-centered: changes across time in the mean scores obtained from an intervention group are compared to those of a control group. In applied settings, unplanned extraneous variables and inconsistent intervention delivery protocols complicate evaluation [4, 5]. Researchers may be able to randomly assign participants to a training intervention or wait list control. However, this makes it difficult to analyse the effects of variables such as age, gender, or indeed prior experiences [6, 7]. Variable-centered tests of difference can have low statistical power due to low sample sizes, attrition and participant heterogeneity [8]. This has contributed to an inadequate pipeline of field research opportunities in psychology and an over-reliance on student populations for research [4]. Moreover, similar group level averages may mask significant individual level differences [9].

Person-centered evaluation involves the identification of homogenous sub-populations within an intervention group so that heterogeneous effects can be identified [10]. Sub-populations can be those with similar pre-intervention characteristics, similar experiences of the intervention, or those reporting similar intervention effects. This approach also requires rare intervention conditions: large populations with stable membership over time within each sub-group. Person-specific evaluation, in contrast, involves detailed exploration of individual journeys throughout an intervention [11]. Person-specific approaches offer promise in identifying how heterogeneous or idiosyncratic experiences and circumstances impact on intervention outcomes. It does not require a large sample size when conducted qualitatively. However, the focus on individual experiences leaves unanswered the basic question of relevance for practitioners and commissioners of interventions: does the intervention work in some way for enough people for it to be considered beneficial [11]?

## Complex field study settings

Applied psychology requires careful selection and recommendation of evidence-informed interventions to address complex psycho-social problems [12]. Practitioners rely on academic psychology to support advice provided to commissioners. The study we present herein provides a good example of a real-life problem, using longitudinal data from a client-led coaching program for dyslexic adults designed to improve workplace performance and reduce risk of unemployment for a marginalized, vulnerable population. Dyslexia is neurodevelopmental condition associated with difficulties in literacy, self-organisation and working memory [13] but also with strong people skills [14] and entrepreneurial flair [15, 16]. Currently, this significant minority (around 8% of the working age population [17]) is overrepresented in unemployment (around 30%) and incarceration (50%+) [18, 19]. There are legislative obligations

for employers to provide disability accommodation interventions in most developed economies [20–22]. The emergence of a Neurodiversity Movement has encouraged employers to increase representation of dyslexia, along with related conditions such as Attention Deficit and Hyperactivity Disorder and autism [23, 24]. Disability coaching interventions are thus increasingly provided as a reasonable accommodation across the world to support the growing need to support visible and invisible disabilities at work [25, 26] and in order to make informed funding decisions, we need to know the extent to which they are likely to work. For example, in the UK the government provides direct funding for accommodations for around 30,000 disabled and 3,000 dyslexic individuals per year [27]. These interventions require significant resources (time, effort and expertise, average $1000 for each dyslexic intervention [28]) from those involved but are rarely evaluated and therefore the implications for employers on how to comply, commission and deliver relevant support activities for dyslexia and other such 'invisible disabilities' are unclear [29, 30]. Employers, disabled people and psychology practitioners (key stakeholders) continue to ask whether these interventions are working, yet psychological research has heretofore failed to support practice with ecologically valid evaluation [23, 31–33].

Our study orientates us in the very real, ethical, social and economic consequences of failing to conduct effective field studies. The imperative to devise conducive methodology that can identify the value of interventions at person-specific, person centered and variable-centered levels. Our initial research, aiming for high standards in field research, collected data from two pragmatically randomized control field samples ($n = 67$, $n = 52$) with triple blind controls. In both studies there were consistent person-specific indications of improvements in behavioural measures, as shown in S3 and S4 Tables, where group means indicate improvement across self-rated measure of job performance development or use of strategies. Review of raw data per individual for these groups (see S2 and S3 Appendices) demonstrated some large gains for individuals in the intervention groups, but not for the control. Such a pattern indicated that the programs had value in improving inclusion and career progression, but effects were not evident in variable-centered analysis of outcome measures (ANOVA) and consistent person-centered clusters could not be identified across the variables. In this type of intervention, multiple mechanisms and idiosyncratic responses are expected and desirable [34] and so instead of accepting the simple null hypothesis result following the marginal $p$-values at the variable centered-level [35], we set out to devise a structure in which we could apply a dynamic assessment of impact at multiple levels of analysis, each to inform the needs of multiple stakeholders.

We summarize our solution to these problems, the Meta-Impact score, as an aggregation of predefined levels of individual improvement across a number of different measures of a range of plausible intervention effects. Meta-Impact is presented as a prototype for consideration by those seeking to capture within-participant variance during a group-wise comparison across time in order to better understand and quantity the overall effects of applied interventions [6]. We begin with a review of the conduct and variable-centered evaluation of two coaching intervention studies and discuss the potential for error in the results. We provide a step-by-step description of the method for calculating the Meta-Impact score, illustrated with data from both intervention studies. We present the results, which can be analysed using NHST, effect size, odds ratio or Bayesian methods. We conclude with a discussion of the strengths and weaknesses of Meta-Impact and its implications for practice.

## Original study context and design

The objective of the intervention research was to provide overdue practice evaluation by ascertaining both if and how a coaching intervention could improve work performance for dyslexic

adults. Before the present study, pilot research using longitudinal dyad data from dyslexic coachees and their respective workplace supervisors provided some evidence that both parties perceived improved work performance [36]. The results of a systemic narrative review of the literature related to improving contextualized work performance via coaching also suggested support for the practice [37] which we now briefly describe in order to explain the potential for individualized pathways in intervention progress.

**Mechanisms for improvement.** An expert consultation process undertaken as part of the narrative systematic review protocol [37] highlighted that several cognitive and psychosocial mechanisms are specifically relevant to improving the work performance of dyslexic adults. These could be considered the basis for an intervention to 'work', as per the personalized pathways predicted in applied psychology. Specifically, working memory (WM, in particular, the capacity to hold and work with information in our attention at any one time [38]) and 'self-efficacy' (SE, our belief in our own ability to act; [39]) were highlighted as potential mediators of intervention success. Coaching interventions have been demonstrated to affect a range of cognitive, behavioural and emotional domains, they are flexible to individual need [40, 41] and are in common use as a disability accommodation [42]. Therefore, a systematic narrative review of 24 coaching interventions was conducted, which revealed four potential domain mechanisms triggered and maintained through the coaching intervention. These were behavioural (implementation of strategies to manage WM-type difficulties), emotional (stress management to reduce load on WM), psychosocial (appreciation of own ability and skill in context, SE) and / or cognitive (WM improvement itself). The review further identified a combination of Social Cognitive Learning Theory [39] and Goal Setting Theory [43] as the basis for a coaching protocol in which four mechanisms could be activated to successfully improve work performance. Social Cognitive Learning Theory contains four main stages: SCLT1: verbal persuasion; SCLT2: vicarious learning; SCLT3: role-modelling; SCLT4: mastery. Goal Setting Theory contains three elements: GST1: goal clarity; GST2: socially contextualized goals; GST3: goal achievement self-efficacy. S1 Fig reproduced from Doyle & McDowall [37], depicts the hypothesized pathways.

When devising the present studies, we originally hypothesized that changes in the scores pertaining to each mechanism would correlate with each other and improve significantly more in the experimental group than those for participants in control groups. However, it is also plausible that each participant may experience one, some, or all of these changes based on their initial needs and the type of coachee-coach interactions taking place within the intervention. In line with a coaching psychology pedagogy, the pace and examples within the coaching intervention were matched to the group needs, balanced across the individuals within each group, however, the topics and exercises chosen as foci remained constant. We conducted two field studies to test the effects. Coaching study 1 (CS1) was undertaken in a high-fidelity field setting (actual work context), followed by Coaching Study 2 (CS2) which was conducted in a controlled quasi-experimental setting (e.g. holding the coaching delivery constant). A third-party review of coaching activities was undertaken to assess the level of consistency and ensure all topics were present for all coaching clients. An outline of the coaching activities which were consistent across coaching clients and experimental groups is provided in S1 Appendix. The study was reviewed and approved by the Ethics Board of City, University of London PSYETH (R/F) 15/16 245. Written consent was obtained for all participants.

Our study investigated the potential confounding influence of individual effects being masked by the use of group mean analysis, and to identify a potential alternative method which could answer questions of intervention effectiveness at the group level, whilst incorporating personalized pathways of change at the individual level. We note that personalized pathways are gathering recognition in human sciences more broadly [44], and hope to contribute

to the development of data-driven, individualized methods to avoid overreliance on observational / qualitative adaptations in these endeavours [45]. We posit that individuals are likely to react differentially even where a standard protocol intervention has been delivered. This requires sophisticated within person and between person/ between group analysis to detect actual patterns of change.

**Study design.** Coaching Study 1 (CS1) was conducted with a UK local government department over a period of 18 months allowing for three waves of coaching and assessment of outcome measures; a necessary concession to ensure that the department was adequately staffed during the intervention. Analysis of group means using ANOVA, including a control variable of job satisfaction, were performed to ensure that the waves were not significantly affected by extraneous temporal effects. There were three coaching conditions (the independent variable): one-to-one coaching, group coaching and waitlist control. Data on outcome measures were collected at three intervals: before (T1), immediately after (T2) and 3 months after coaching completion (T3). The outcome measures used are shown in S1 Table (measure description, scoring, sample item and reliability data for each in S5 Appendix), comprising six dependent variable sets at three intervals. Initial results showed a mixed pattern of intervention effects.

The intervention was repeated as a quasi-experiment, with refinements to measures and protocol to ascertain if the mixed results were due to artefacts (such as the use of general self-efficacy, where work-specific was needed) or if the results would be replicated in a new sample (for example CS1 took place in one organization, it is possible that the results pertained to organizational support rather than the intervention itself). Coaching Study 2 (CS2) mirrored the intervention design of CS1 but, for practical reasons of controlling the protocol, cost and time, had only two independent variable conditions (the group coaching and a waitlist control). In CS2 more sensitive questionnaire measures than those used in CS1 were selected to examine whether the results of CS1 were impacted by Type II error, but these remained conceptually aligned to the four domains using seven outcome measures at all three measurement points. S1 Table provides an overview of the measures per domain (for more details see S4 Appendix). The large number of outcome measures meant that even when Bonferroni correction was minimized, significant results could be masked [8]. Despite careful preparation of study design, including a priori power analysis and accordingly targeted sample size, the number of variables to be analysed and participant attrition resulted in low observed post-hoc power and borderline *p*-values, as presented in S4 Table.

Triple blind controls were implemented in both studies. The researcher received coded data via an administrative assistant and so was unaware of the intervention delivered to each participant; the cognitive tester was not informed whether participants were intervention or control and the coach was not aware of participants' pre-intervention scores. Pragmatic allocation to the conditions [26] meant that groups were initially randomized but individual participants were then allowed to indicate if they had an urgent need for intervention or, in CS1, a strong reason for needing to be part of either group or one-to-one intervention. Seven people from CS1 and four people from CS2 moved from control to group coaching to accommodate such requests. These individuals do not contribute unduly to the overall effect: there were no significant differences in pre- or post-group mean scores when comparing means with and without these participants included, though they do contribute individual Meta-Impact scores. Though this is a potential confounding influence on the results, it reflects the real-world problems experienced in human sciences when restriction of intervention could compromise participant well-being [46]. This is an issue of research ethics and integrity as much as of principles of controlled design.

**Sample size.** Initial sample sizes were CS1 *n* = 67 and CS2, *n* = 52, as determined through a priori power analysis using a prediction of medium effect sizes (as determined from the

systematic review). Not all participants completed, and some presented for cognitive testing at all three intervals but failed to return questionnaires (or vice versa). Variations in self-organisation skills within dyslexic populations may account for this. Therefore, the sample sizes at T2 and T3 intervals varied, ranging from $n = 32$ to $n = 46$ for some dependent variables in CS1, and were reduced to $n = 46$ for some in CS2.

## Results of the initial variable-centered evaluation

S2 Table (CS1) and S3 Table (CS2) show the means and standard deviations for all dependent variables, per time interval, per condition for CS1. Post-hoc observed power ranged from $\beta = 1.0$ to as low as $\beta = .077$. We adjusted the analysis include both parametric and non-parametric group x time analyses (CS1:3 x 3; CS2: 2 x 3). S4 Table shows the sample sizes, analysis method, power, significance and effect sizes for an example measure from each domain, to illustrate the pattern of results. Though multiple analyses were performed, we have selected to report mainly RM ANOVA time and groups in order to provide consistency for comparison across the variables, with one exception necessitated by variation in T1 scores. A MANOVA presentation was considered but the potential for individual differences to result in idiosyncratic changes led us to focus on identifying change in each outcome variable.

As shown in S2 and S3 Tables, the intervention group means showed a positive trajectory, with some medium effect sizes as shown in S4 Table. The group-wise variable-centered comparison produced significant $p$-values for only two variables in CS1 (Behaviour and SE) and only one in CS2 (SE) which did not achieve significance with the Bonferroni correction applied. We also tested whether there could be an overall effect of the intervention that was evident across different highly reliable measures (all questionnaires exceeded the standard .7 reliability coefficient, both in original publication and in our samples). To do this we calculated an improvement score for each variable (T3 score minus T1 score) and examined the correlations between them. Few significant correlations were observed between the domains in either study and even those with a significant $p$-value were not large (ranged from $r = 0.042$ to $r = 0.556$; see S5 and S6 Tables). This indicates that the difference over time on one outcome does not allow us to make simple and reliable predictions about the difference over time on another outcome.

## Interim discussion

Using traditional NHST we thus observed limited evidence of intervention effects from the variable-centered evaluation from CS1. This was also evident in CS2 despite the use of improved measures. Qualitative feedback from participants, managers and coaches however indicated that they found the coaching intervention beneficial to their work performance. This was similar to reports from the pilot research [36]. Only three participants dropped out of the intervention in CS1 and one in CS2; such low attrition is congruent with verbal reports in that they valued the experience sufficiently to give of their time. The low correlations between improvement scores (S5 and S6 Tables) provided some preliminary evidence that variable-centered results could be impacted by idiosyncratic intervention effects.

We considered whether the interplay between multiple means, observed power, significant and effect size was a true null result [47] or was indicative of a lack of homogeneity in response to treatment amongst individual participants. Given the poor alpha values, the effect sizes identified are unexpectedly medium to large when compared with general results in applied intervention studies [48], which caused us to consider whether we had missed palpable improvements at the individual-level. An outcome supporting the null hypothesis may have been unduly influenced by group-level aggregation. Specifically, the variable-centered

approach is not sensitive to individual-level variability in trajectories from pre- to post-intervention scores, particularly when multiple variables require stringent corrections to accepted *p*-value cut offs in significance as remains traditional practice in applied psychological research [8]. To summarize, we argue that NHST is not suitable in in these circumstances not solely because of the arbitrary dichotomization of *p*-value, but because the group-mean comparison has masked the between-participant variability in the intervention pathways. This error requires more than a recalibration using new statistical approaches such as confidence intervals and effect sizes [49] it requires a fundamental rethink of the group means comparison, per variable, as the standard marker of success.

To better understand these individual differences in variable-specific trajectories, the magnitude of the effect from T1 to T3 scores, per outcome variable, per individual was reviewed. Observation of raw improvement scores in both CS1 and CS2 indicated a clear pattern; the reason that the correlations between improvements across different outcomes were low was because the people improving in each variable were different people. Specifically, some improved on cognitive measures but not behavioural, some on emotion measures but not cognitive measures etc. This pattern, which was masked by variable-centered analysis, was found in both studies.

Our finding provided a rationale for using person-centered analysis. An exploratory cluster analysis using *k*-means was initially conducted on the cognitive measure. Within the intervention groups for CS1 and CS2, there were two distinct clusters: those who were comparable to the control group in terms of the amount of change (CS1, *n* = 12; CS2, *n* = 12); and those who improved a great deal (CS1, *n* = 18; CS2, *n* = 14). Those improving measured far in excess of the lift associated with practice effects as reported by the test publishers [50]. Some people from the intervention group improved and some did not. Few from the control group improved beyond practice effects, as the results below will indicate, in CS1 58% of the intervention group reported at least one improvement, compared to only 8% of the control group. In CS2, 69% of the intervention group report at least one improvement, compared to 35% of the control group. These different trajectories suggested that we were working with sub-groups and a person-centered approach would be needed to evaluate the intervention. S1 Graph depicts the means at each interval for the CS1 cognitive ability scores per cluster. As shown, those who did not improve show a similar pattern of response to intervention as the control group.

The person-centered analysis did not explain the variance in the behavioural, emotional and SE-based variables; no reliable clusters could be found. Instead, there was evidence of person-*specific* responses to treatment [11], with a more evenly graduated set of improvement magnitudes across the groups, though again the highest improvement scores were predominantly from the intervention groups. This complexity and richness of intervention trajectories indicated the need for a person-specific approach. We formed an abductive hypothesis that the intervention was having a measurable effect for most intervention participants, but the effect was inconsistent across the potential variable pathways. Therefore, all our mechanism pathways were relevant to some participants, but not to all.

## Introduction to the Meta-Impact score

The first author established a quantitative method for evaluating person-specific responses to treatment in order to answer the question: "did it work?" at the individual level. Participants showing a significant improvement with *any* of the domain variables were identified, using a demarcation as described in the method below. This was based on assessing the numerical point at which a 'Real World Value' is most likely to have occurred. Real World Value is

defined here as a criterion-validity approach, the point of improvement after which palpable differences in performance are likely to be evident, transferred back to the workplace, and then improving the dyslexic individual's job retention and longer-term career progression. We note that there exists a far wider debate on the extent to which a numerical data point reflects an authentic psychological experience, however such a debate is more fundamental to the human and social sciences, exceeding the remit of this paper. See research concerning the 'Quantified Self' for contemporary exploration of this conundrum [51, 52]. The number of intervention participants reporting any significant impact was then compared to the number of control group participants reporting any significant impact, a layering of individual then group-wise comparison. Together, these two main demarcations (significant gain in any variable, per individual and the frequency of gain per group) form the Meta-Impact analysis. The purpose of this new analytic method was to consider whether the intervention was associated with observable change at the group level, albeit by working differently for each person, thus combining the advantages of person-specific and variable-centered evaluation. We formulated this new set of exploratory hypotheses to guide the analysis:

H1$_a$ Coaching participants are more likely to have experienced one or more significant, positive improvement on intervention-relevant variables compared to control group participants.

H1$_b$ There will be a significant difference between intervention group participants and control group participants regarding the number of variables in which a participant reports experiencing a strong improvement.

We choose these hypotheses deliberately as markers of 'Real World Value'. The answers to these questions will support a customer-centric approach [53] to statistics, helping stakeholders to decide whether or not coaching can be recommended as a disability accommodation.

## Method

A new data set was created with the following four steps: (1) calculating an improvement score for each participant for each outcome; (2) converting the improvement score to a standardized *z*-score to remove differences in the numerical range of the measures; (3) determining what magnitude of z-score constitutes a real-world improvement and (4) computing a 'Meta-Impact' score for each participant, forming a new datum from which Meta-Impact mean per treatment condition, variable or participant could be calculated. Each step is now outlined in detail.

### Computing the scores and improvement range

**1. The raw change.** As above and as in the original analysis, this is based on the T3 'after' score minus the T1 baseline score. Difference scores have received critique from a pure empiricism standpoint [54] but have important qualities that give them utility when calculating Meta-Impact. Difference scores retain value from a rational empirical perspective within the context of applied intervention research as they can be used to examine the likelihood that an individual has reported change of a magnitude that is linked to meaningful behavioural outcomes. This helps to answer the question of central importance to stakeholders: did it work? Change experienced during an intervention or immediately afterwards, as in T2, is likely to be influenced by the presence of the practitioners (Hawthorne Effect [55]) and as such the intermediate scores offer little practical value. We have focused our interest on the retained improvement, and therefore the final T3 represents not change experienced during the intervention but change sustained after the intervention has ceased and normal work has resumed.

From the perspective of those commissioning interventions, there is more value in the magnitude of this effect than of understanding the curve or progression towards, since this final score will determine whether or not the individual will perform their job at an improved level and thus retain or progress within their employment.

The raw change, $M_{n,m}$, for each outcome.

$$M_{n,m} = (T_f - T_i)_{n,m}$$

where

$T_i$ = the initial, pre-intervention, score.

$T_f$ = the final, post-intervention, score.

**2. Standardizing the numerical range of improvement.** A standardized $z$-score was calculated for each improvement score for each individual for each dependent variable (i.e. the number of standard deviations from the mean improvement score). This was necessary to create a consistent format for evaluation as the original scales used a combination of scaled/interval data ranging 1–19 (cognitive) and 1–5 or 0–3 (Likert scales in the questionnaires).

The mean and standard deviation of the raw change, $M_{n,m}$, is calculated for each variable m.

The Mean for each variable m is $\mu_m$.

$$\mu_m = \sum_{k=1}^{n} \frac{M_{n,m}}{n}$$

The standard deviation for each variable m is $\sigma_m$.

$$\sigma_m = \sqrt{\frac{\sum_{k=1}^{n}(M_{n,m} - \mu_m)}{n}}$$

The standardized z-score for each variable m for each participant n is $z_{n,m}$.

$$z_{n,m} = \frac{M_{n,m}}{\sigma_m} - \mu_m$$

**3. Determining the margin of improvement.** Small improvements due to measurement errors, potential practice or random effects were eliminated by restricting the number of people in each variable who scored as improved. A numerical limit of *comparative* improvement score was set at least plus one standard deviation from the mean improvement (improvement $z$-score of = />+1). We propose that our limit represents a numerically distinct improvement in *actual* score that given the validity of the measures used is linked to meaningful change in terms of post-intervention thinking and behaviour. Thus, this delineated a group of significant improvers, who were likely to experience a tangible improvement following intervention.

The group of participants deemed to have demonstrated a significant improvement is determined by applying a condition that the improvement, as shown by the z-score, is greater than, or equal to, 1 standard deviation.

*if $z_{n,m} \geq 1$ then improvement is deemed significant.*

*Note*: *Defending a dichotomization.* In order to defend the use of dichotomization, we must link its use to the tangible effect it creates in 'Real World Value'. In this section we describe the logic of how the limit was contrived for our sample, in the discussion we will debate the principles for use more broadly. We use the cognitive scores (WM) as the example here since they

are particularly well-evidenced in terms of practice effect margins with reliable standardized samples [50]. The average improvement for WM across all conditions and studies was +1.03 of a standard score (the standard scores range from 1–19; the average range is 8–12). CS1 and CS2 improvement from T1-T3 for the intervention groups was an *actual value* change in standard scores of between +0.98 and +1.81, within the expected bands of a standard deviation for WM scores but in excess of the typical practice effect of 0.6 reported by test publishers [50, 56]. Following conversion of the raw change improvement magnitudes to *z*-scores in the data sets, $z = 0$ would therefore represent a cognitive ability improvement of +1.03, higher than a practice effect and similar to improvements reported as significant in computerized working memory training research [57, 58]. However, WM training has been widely criticized for failing to demonstrate real world value; small increases in WM do not transfer to contextual performance and lack ecological validity [59]. Thus, we surmised that we needed to raise the bar higher than the average improvement in our data set. Data from both CS1 and CS2 indicated that a *comparative* value *z*-score = />1 represented an *actual* value standard WM score increase of = />2.5, which then is far in excess of an accepted 'working' effect and is more likely to represent a real, palpable change for the participants with implications for them beyond the testing environment. Taking the same approach through the remaining variables, analysis of means and standard deviations indicated that the actual change in score represented by *z*-score of = />+1 was ranged from 0.28–1.7, using Likert scales with four or five intervals (see S4 Appendix).

To summarize, in order to qualify as tangibly improved on any given variable in this analysis, a participant had reported a comparative improvement score of $z = />1$. Given the criterion-related validity of the measures used, we propose that this drew out those who had experienced palpable change while reducing the risk of within-person common methods bias [60].

**4. Meta-Impact score.** Finally, each participant received a new improvement score: 'one' for each variable which registered an improvement *z*-score = />1, and 'zero' for z-score<1. Note that at the individual level of analysis, the presence of a score of = />1, indicates success as opposed to failure, and so for person-specific research this is a reasonable end point. However, in order to assess success at the group level, it is necessary to understand whether an intervention group member is more likely to experience success than a control group member and, as such we then combine improvement scores to assess the group level frequency of improvement.

A new scoring system is now imposed for each participant.

$$If \ \ z_{n,m} \geq 1 \ \ then \ \ MI_{n,m} = 1 \ \ otherwise \ \ MI_{n,m} = 0$$

An MI score, $MI_{\mu n}$ is now calculated for each participant, which can also be converted to a mean MI score for comparison across studies.

$$MI_{\mu n} = \frac{\sum_{k=1}^{m} MI_{n,m}}{m}$$

To demonstrate, the average Meta-Impact (MI) scores per treatment condition was then calculated; the higher the score, the more likely that participants experienced at least one significant change. Mean *z*-scores and mean MI scores per condition are shown in S7 and S8 Tables for CS1 and CS2 respectively. Individual *z*-scores and MI scores are shown in S2 Appendix (CS1) and S3 Appendices (CS2) for all participants.

## Data analysis

Individual MI scores were then used as the reference variable for group-wise comparisons to investigate the impact of the interventions in CS1 and CS2 according to the abductive hypothesis 1. Group-wise comparisons of the new variable were planned using non-parametric tests selected according to the number of independent groups. We calculated post hoc effect size *r* for non-parametric comparisons to avoid sole reliance on the *p*-value significance and considered these in relation to impact magnitude in reported applied psychology more generally to assess value [48]. This addressed the question 'did it work?' from the employer, or intervention commissioner perspective, as well as aiding the practitioner in reflecting on intervention design and delivery. We also present a per variable summation and conducted a basic odds ratio analysis, to demonstrate the flexibility of the MI data with alternative statistical paradigms.

## Results

### Hypotheses 1: Differences between intervention and control

**CS1 results.** S7 Table shows the descriptive statistics for the mean *z*-score per variable, the group-level MI scores for CS1 and the number of individuals achieving a = /+1 *z*-score per variable. Differentiation between the mean *z*-scores is presented for due diligence and transparency of reporting only and not for further inferential statistical analysis; group-wise comparisons of these scores would yield the same results presented in S4 Table. From a person-specific perspective, the CS1 group coaching participants achieved an average of more than one improvement per person (condition mean MI = 1.13) with ten (out of 15) achieving at least one improvement; the one-to-one participants achieved an average of almost one improvement per person (condition mean MI = 0.88) with eight (out of 16) achieving at least one improvement; and the control group was almost zero (condition mean MI = 0.08) with only one (out of 12) achieving an improvement. S7 Table also demonstrates the contribution of each variable made to the overall group MI, with the number of individuals achieving an MI ranging from one to seven individuals. Self-efficacy did not seem to contribute much to the Meta-Impact (only one improvement) with all other measures improving for between four and seven participants each.

Reverting to a traditional NHST perspective we then conducted a Kruskal-Wallis test for between-group effects on MI scores, which identified a significant difference in MI scores ($K$ (2,43) = 9.379, $p$ = .009). Post hoc Mann Whitney comparisons of the MI score revealed a significant effect for group coaching vs control ($U(27)$ = 2.447, $p$ = .014, $r$ = .47), but not one-to-one versus control ($U(29)$ = 1.873, $p$ = .061, $r$ = 34). Both effect sizes are in the medium-large range based on Cohen's original proposal and compared with recent reviews of typical effect sizes in similar fields of study [48]. It may be that the second comparison's $p$-values are compromised by small sample size and consequential lack of power (as in the original study). S2 Graph shows the MI means and confidence intervals for each condition (1 = one-to-one; 2 = control; 3 = group coaching).

Also appropriate at this point would be to calculate an odds ratio, i.e. the likelihood of an individual improving via coaching depending on group allocation, which would be as follows: group coaching = 10/15 (66%); one-to-one = 8/16 (50%); control = 1/12 (8%). Combining the intervention groups, the chance of the coaching having a positive affect was significant compared to control: $OR$ = 16.62 [$CI$ 1.91–144.24] $z$ = 2.55, $p$ = .011. A Bayesian approach could also be considered for analysis of the Meta-Impact score, however the development of the prior would be dependent on existing subject matter knowledge. In the example of coaching

used as a disability accommodation, we would be reliant on subjective data to determine our prediction, since the field is under-developed and as such we have not presented a Bayes factor analysis herein.

**CS2 results.** S8 Table shows the descriptive statistics for the mean calculated $z$-score per variable, the mean MI scores for CS2 and the number of individuals achieving a = /+1 standard score per variable. Again using a NHST approach we can discern a significant difference between-groups for MI ($U(52) = 3.014$, $p = .003$, $r = .42$). The mean MI score for the intervention group was 1.46, i.e. an average of 1.46 improvements per person. Again, considering an odds ratio approach we note that in the intervention group, eight (out of 26) participants failed to register a single significant improvement across any of the variables (18/26 improved), yet for the control group this was far higher with seventeen people (out of 26) failing to register a single significant improvement (9/26 improved). Thus, in CS2, the likelihood of an intervention group participant significantly improving was higher than control: $OR = 4.25$ [$CI$ 1.33– 13.56] $z = 2.44$, $p = .014$. See S3 Appendix for individual scores; again, there are some missing data sets. S8 Table again demonstrates the contribution of each variable to the overall group MI, indicating that for each, between two (behavioural–strategies) and nine (emotional–stress) individuals experienced an individual improvement.

In S9 Table, we illustrate two case study trajectories for individuals from each coaching study to provide insight into how individuals have contributed to the analysis through their idiosyncratic pathways, but towards a common end of an overall gain from the intervention.

## Discussion

We outline how our revised analytic method allowed us to test the hypotheses to support practical decision making and then discuss study limitations as well as implications for research and practice.

### Control versus intervention groups comparisons

The more commonly used variable-centered, group-wise analysis in the initial studies was impacted by Type II error. Calculating a Meta-Impact score allowed us to apply inferential, frequentist statistics for group-wise comparison analysis, thus retaining the benefits of a variable-centered analysis but with an individualized pathway. Our novel method produced data indicating that we should reject the null hypothesis, specifically that (H1a) coaching participants are more likely to have experienced one or more significant, positive improvements compared to control group participants and; (H1b) there was a significant difference between intervention group participants and control group participants regarding the number of variables in which a participant reports an improvement. The MI result was consistent with verbal reports and less subject to reduced statistical power caused by attrition, heterogeneity in intervention experiences and response to treatment, or individual differences in baseline performance.

Consistent with the original analyses, the group-wise MI comparison repeats the observation that the group condition outperformed the one-to-one condition: effect sizes were larger ($r = .47$ and $r = .34$ respectively). The one-to-one coaching MI $p$-value alone suggests that it was not a successful intervention for dyslexic adults. However, with a medium effect size and small data sets there is still again risk of a Type II error when we use NHST. The MI score comparison may mitigate, but it does not negate the need for sample size consideration in field studies, or overcome consideration of alternative methods using a balance of effect size calculations or a Bayesian approach [1, 35]. An interesting perspective for practitioners at this point may be to highlight which individuals have improved and which have not, thus indicating a need to provide additional, alternative interventions for this subgroup. As such, the different

levels of analysis provided by the Meta-Impact score can provide useful data for practitioners in applied field settings.

## Study limitations

**Dichotomization of variables.** Dichotomization of variables (for example $p$-value cut-offs, effect size boundaries) has been critiqued as an unnecessary step which sacrifices numerical nuance and specificity [61]. The American Statistical Association clearly advises that policy decisions should not be made on $p$-values alone [35], yet most intervention evaluation studies still choose their dependent variables a priori and place a simple 'yes' or 'no' result on the success of the intervention to achieve change in each of the measures, which typically represents only one domain [62]. However, in the eyes of the participant, the practitioner and those responsible for funding the interventions, some form of evidence-based dichotomization is needed to answer the question: did it work? Our improvement score delineation facilitates nuance at the group level by sacrificing it with dichotomization at the individual level. Both activities are aimed at increasing psychological understanding of intervention impact. A binary yes/no at the individual level helps us circle back to support vulnerable individuals who may need further help, whose experiences are lost in a group average and therefore provides the real-world value for realist researchers in field settings. The MI score provides a more nuanced answer at the group level, using $p$-values, effect size, odds-ratio or Bayesian approaches to determine likelihood of positive investment return for commissioners of interventions.

Throughout psychology dichotomization has been used to pragmatically support assessment of impact and that delineation persists despite critique in order to support decision-making in practice. Although acknowledged, the fundamental resolution of this limitation falls outside the remit of this methods-focused article. Yet with this in mind, Meta-Impact has included stringent criteria for success; the dichotomization makes rejection of the null hypothesis dependent on the reporting of meaningful changes. Meta-Impact is intended to reduce the risk of type II errors due to group-wise variable-centered masking of individual nuance. Dichotomization is according to practically relevant criteria, which mitigates a corresponding Type I error risk. The margin of improvement demarcation enables group analysis of whether participants improved significantly on any measure, rather than (a) all measures or (b) cumulative practice effects across all measures. It links numerical patterns to perceived intervention value, a pragmatic concession bridging person-specific, centered and variable centered designs in applied research with modest sample sizes. Further investigation is needed on this point. The logic of $>/ = +1$ SD may not apply to all variables in all contexts, in some studies $>/ = +0.5$ SD, or $+2$ SD may be more appropriate to identify the real-world impact. We recommend mixed method studies to cross reference qualitative experiential reports with numerical value as a future development in honing the method.

**Novel technique.** The goal here is not to replace multi-variate analysis of variance or more complex path analysis, but to challenge the group-wise comparison premise in realistic settings where individuals react differently to the same intervention and sample sizes are frequently compromised. Alternatives were considered: as a comparison with traditional methods, we also tested a cumulative total of effect magnitude per participant to smooth out the personalized pathway into a whole intervention effect. Significant differences were found between intervention and control group measures (CS1 $F(2,22) = 6.868$ $p = .005$; CS2 $t(42) = 2.193$ $p = .034$). While this analysis further supports the contention that the intervention group experienced an effect from the coaching and will inform decision makers, it does not help us to understand the different routes that each participant may take, and the extent to which each variable contributed to the overall impact. In other words, it provides outcome evaluation

without process evaluation. As such it provides little value to practitioners seeking to hone and understand intervention success by understanding participant journeys and value of different mechanisms. Identifying improvement scores for each individual for each variable provided important process evaluation data in our study. We suggest that further research is needed to explore utility of the method with different hypothesized outcomes. We hope that we will stimulate further debate about overuse of group means to predict and prescribe nuances in human behaviour.

**Null hypothesis testing.** One could also argue that the very premise of null hypothesis testing across independent groups has limited practical value in this context, and that experience sampling methods would have been more appropriate for determining within-person variance [63]. Such approaches do not meet the needs of a customer-centric approach [53] which, as stated, often requires evidence from between-groups comparisons in order to make decisions proceed with an intervention investment. The authors accept the nuances of the New Statistics movement and the critique of any dichotomization in research methods [49]. However, we argue that in practice, all decisions boil down to a binary. Did it work? Shall I participate? Shall we invest in this intervention? As such, in order to remain relevant to practice, psychologists must offer guidance to stakeholders and as such more definitive methods will always be relevant to applied settings.

Historically, evidence-informed practice has relied on primary studies using the group-level analysis such as (M)ANOVA and multiple regression, which risks Type II error in field research. In support of the need to move beyond comparison of group means as the dominant method we argue that, even when successful, the approach leads to one-size-fits-all interventions. These may then be overused by delivering them to some participants for whom they are ineffective. A good example of this is mindfulness: the true extent of heterogeneity in response remained hidden by group-level analysis and a vulnerable minority subset may have been exposed to the intervention [64, 65]. In Meta-Impact, we are expanding customer-centric research [53] to include the commissioner, recipient and the practitioner delivering the intervention as stakeholders. We propose an accessible method that accommodates the needs of all three. The Meta-Impact calculation process we have described progressively layers the analysis to meet the needs of stakeholders: the individual MI (recipient) to the group MI (commissioner), then supporting the practitioner in determining 'how did it work' within the overall result by capturing the contribution of each variable. This is easily applied in professional practice, where smaller sample sizes are common-place, and we need to avoid reliance on cross-sectional studies.

## Implications for research and practice

The main implications for research are twofold: Firstly, the development of a data analysis strategy that accommodates diverse responses to interventions and secondly, the finding that participants in a coaching intervention are potentially supported through coaching psychology interventions.

The Meta-Impact analysis contributes to our understanding of how a complex intervention works at person-specific level in real-life settings and, as a prototype, provides a method for application to field research. Between-persons, time-series studies are hard to conduct in management science [66], are lacking in applied psychology research [67] and specifically within dyslexia accommodation research [16, 17, 68, 69]. Study designs aimed at closing this gap are known to be challenging given lack of control over extraneous, heterogeneous life events [5, 70, 71]. Traditionally, these have been documented and quantified as control variables, which has the side effect of then increasing the multiple required for a Bonferroni correction and

reducing statistical power. We propose the MI methodology as a pragmatic solution to allow intervention effects on participants to be understood and isolated. It can be used to identify those differentially impacted by environmental factors and individual differences as it permits us to collate multiple dependent variables in one study without over sacrificing power. Though our participant trajectories lacked uniformity in the mechanism of change, each is still able to contribute an outcome of change to a variable-centered group comparison evaluation which allowed us to gain a sense of the intervention effectiveness. Attempts to hone interventions down to constituent homogenous 'active ingredients' appear to be limited because of person-specific effects. In applied psychology settings, commissioners benefit from a customer-centric approach [53], knowing what intervention *has a good chance of working in some way for many* rather than the one mechanism that *will often work in the same way for many*. The knowledge drawn from the Meta-Impact analysis can be used to provide clear guidance to employers, whilst practitioners can employ the detailed analysis to personalize delivery protocols, so that fewer participants experience a null or adverse effect. Our method facilitates the ecological validity of the field setting with the rigor of a realist empirical analysis.

Secondly, we note the implications and relevance of the original studies' aims. We set out to discover how providing coaching for dyslexic employees might support their work performance and improve the sustainability and progress of their careers. In doing so, we selected variables previously highlighted as adversely impacted by dyslexia and likely causes of exclusion. In both coaching interventions, and particularly group coaching interventions, were able to improve participant outcomes in these mechanisms, using a well-constructed, comprehensive coaching psychology approach. Of particular note is the potential impact to working memory, and working memory related performance, which affects not only those diagnosed with dyslexia but also those with Attention Deficit and Hyperactivity Disorder [72], Developmental Coordination Disorder [73], Tourette Syndrome [74], Acquired Brain Injury [75], Multiple Sclerosis [76] and many more hidden disabilities. Working memory research has heretofore focused on improvement via isolated, de-contextualized brain-training games and has been criticized for lacking ecological validity within the social-cognitive development of ability and applied skill [37, 59, 77]. This research indicates that performance difficulties related to working memory, and executive functions more broadly, may be susceptible to the personalized, socially-contextualized development that coaching can provide as suggested by previous research [37]. In order to protect employment and address the significant, worldwide disability employment gap [78] it is essential for applied psychology practitioners to develop evidence-based interventions and supports, and to evaluate the effectiveness of these in situ. Our study is thus also intended to inspire a new direction for performance-related research and disability inclusion work. In an aging and increasingly neurodivergent workforce, applied psychological science can no longer afford to ignore the disabled population [33, 79]. Our method offers practical guidance to psychological researchers and practitioners alike seeking to improve the evidence-base upon which we can inform the services we offer as a profession.

## Conclusion

We argue that for complex, unpredictable and difficult-to-control intervention settings, researchers could use the Meta-Impact analysis to gather data on several 'possible' variables to maximize the opportunities given to us by organizations and working participants where attrition and heterogeneity of response-to-treatment can hinder research; for example a multi-factorial cultural change program involving training knowledge transfer, attitude change, changes to structures and/or process. We propose that Meta-Impact aligns with a shifting realist paradigm and hope that our proposed method can be refined, leading to a wider discussion

on the dynamics of individual and group level analysis. As personalized treatment pathways become more popular in medicine, clinical work and applied fields generally, we hope this gives researchers some ideas on balancing the competing priorities when conducting research that will drive policy. The prototype MI score is an attempt to retain the merits from person-centered, person-specific and variable-centered designs and combining them into one result, enabling us to consider the effectiveness of an applied psychology practice which is delivered widely and lacking in evaluation. We are not the first authors to conclude that participants respond heterogeneously to treatments; we propose that we may be the first to capture these variances using quantitative, empirical logic in order to evaluate the impact of change programs between intervention and control, particularly for individualized treatments such as coaching.

## Supporting information

**S1 Appendix. Coaching content protocol for one-to-one and group conditions.**
(DOCX)

**S2 Appendix. Z-scores and MI score per participant for CS1.**
(DOCX)

**S3 Appendix. Z-scores and MI score per participant for CS2.**
(DOCX)

**S4 Appendix. Actual score cut off points delineated by the $>$/ = +1 SD dichotomization.**
(DOCX)

**S5 Appendix. Sample items and internal consistency analyses for the dependent variables.**
(DOCX)

**S1 Fig. Hypothesized pathway of context, intervention, mechanisms and outcomes for dyslexia coaching [37].**
(TIF)

**S1 Graph. Working memory standard scores per group per time interval, CS1.**
(DOCX)

**S2 Graph. Meta-Impact means and confidence intervals for CS1.**
(DOCX)

**S1 Table. Study domains and outcome measures.**
(DOCX)

**S2 Table. Means and standard deviations for CS1.**
(DOCX)

**S3 Table. Means and standard deviations for CS2.**
(DOCX)

**S4 Table. Group and time-wise comparisons for example variable from each study domain.**
(DOCX)

**S5 Table. Correlational analysis for CS1 raw changes.**
(DOCX)

**S6 Table. Correlation analysis for CS2 raw changes.**
(DOCX)

**S7 Table. Descriptive z-score statistics and Meta-Impact scores, CS1.**
(DOCX)

**S8 Table. Descriptive z-score statistics and Meta-Impact scores, CS2.**
(DOCX)

**S9 Table. Person-specific examples from each case study.**
(DOCX)

## Author Contributions

**Conceptualization:** Nancy Elizabeth Doyle, Almuth McDowall.

**Data curation:** Nancy Elizabeth Doyle.

**Formal analysis:** Nancy Elizabeth Doyle.

**Funding acquisition:** Nancy Elizabeth Doyle.

**Investigation:** Nancy Elizabeth Doyle, Almuth McDowall.

**Methodology:** Nancy Elizabeth Doyle.

**Project administration:** Nancy Elizabeth Doyle.

**Resources:** Nancy Elizabeth Doyle.

**Software:** Nancy Elizabeth Doyle.

**Supervision:** Almuth McDowall.

**Validation:** Almuth McDowall, Raymond Randall.

**Visualization:** Nancy Elizabeth Doyle, Almuth McDowall, Kate Knight.

**Writing – original draft:** Nancy Elizabeth Doyle, Almuth McDowall.

**Writing – review & editing:** Nancy Elizabeth Doyle, Almuth McDowall, Raymond Randall, Kate Knight.

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
