## [Decision Letter · Decision Letter 0]

13 Dec 2021

PONE-D-21-08221Does it work? Using a Meta-Impact score to examine global effects in quasi-experimental intervention studies.PLOS ONE

Dear Dr. Doyle,

Thank you for submitting your manuscript to PLOS ONE. After careful consideration, we feel that it has merit but does not fully meet PLOS ONE’s publication criteria as it currently stands. Therefore, we invite you to submit a revised version of the manuscript that addresses the points raised during the review process.

Please see the reviews below. The reviewers have pointed out several areas where the motivations and arguments could be strengthened, and requested further information where some uncertainty remains.

We look forward to receiving your revised manuscript.

Kind regards,

Hanna Landenmark

Senior Editor

PLOS ONE

Journal Requirements:

3. Thank you for including your ethics statement:  "The study was reviewed by the Ethics Board of City, University of London PSYETH (R/F) 15/16 245

Consent was provided in written form for all participants".   

Please amend your current ethics statement to confirm that your named institutional review board or ethics committee specifically approved this study. 

"I have read the journal's policy and the authors of this manuscript have the following competing interests: The lead author is employed as a director by a non-profit entity who provides services for adult dyslexics."

6. Please amend your list of authors on the manuscript to ensure that each author is linked to an affiliation. Authors’ affiliations should reflect the institution where the work was done (if authors moved subsequently, you can also list the new affiliation stating “current affiliation:….” as necessary).’

Reviewers' comments:

Reviewer's Responses to Questions

**Comments to the Author**

1. Is the manuscript technically sound, and do the data support the conclusions?

Reviewer #1: Yes

Reviewer #2: Partly

2. Has the statistical analysis been performed appropriately and rigorously? 

Reviewer #1: Yes

Reviewer #2: N/A

3. Have the authors made all data underlying the findings in their manuscript fully available?

Reviewer #1: Yes

Reviewer #2: Yes

4. Is the manuscript presented in an intelligible fashion and written in standard English?

Reviewer #1: Yes

Reviewer #2: Yes

5. Review Comments to the Author

Reviewer #1: Thank you for giving me the opportunity to review this paper. It offers a well-constructed and supported rationale for taking this innovative approach to the analysis and interpretation of data in real-world studies. As such, it is certainly needed and this need is underlined by the challenges facing all those seeking to understand ‘what works?’ It is good to see an individual differences type approach to group-level intervention, however I wonder whether there is also an opportunity to use the examples of advances in other fields – such as medicine – as leverage in the argument that work-based interventions should also move towards a more individually tailored approach.

The data presented clearly exemplify the methodological challenges faced in conducting real-world research, as well as the dilemmas in reaching decisions about whether an intervention works/doesn’t work with a client group. The authors are to be congratulated on the efforts taken in gathering and refining these data sets.

A main thrust of the paper appears to be that some participants performed better in some areas than the control group, but there is a claim on page 12 for ‘consistent person-specific indications of improvements in confidence and work-related performance’ in both of the studies described. I suspect it would help support such a statement with clear reference to where this is evidenced and/or what this means in the context of the study, i.e. were these reflected in test results, participant feedback, etc? This could be important as this is one piece of important evidence on which the premise of the rest of the paper is based.

Due care appears to have been taken in describing the various calculations and so it is the Meta-Impact score which becomes the focus of attention. Described on page 13 as ‘an aggregation of predefined levels of individual improvement across a number of different measures of a range of plausible intervention effects’, it is at least partly based on ‘assessing the numerical point at which a ‘Real World Value’ is most likely to have occurred’. This does pose an interesting question about what this ‘numerical point’ might be and I wonder whether this paper can afford to be bolder in suggesting it holds the key to how this figure might be derived, using the statistical processes described later? Alternatively, I wonder if there is also a risk that the thorny issue of when a difference on paper equates to a difference in behaviour may get in the way of what the paper is hoping to convey. Perhaps signposting other papers who do so, might help at this point?

In terms of the study design, it seems reasonable to have made ‘refinements to measures and protocol to ascertain if the mixed results were due to artefacts or if the results would be replicated in a new sample’, however it does raise the question of what these might be. It is not much of a loose end in itself, but in the later context of moving clients from one group to another, these may be seen as potential ‘liberties’ in the eyes of some readers. Care is taken to state there were ‘no significant differences in pre or post group mean scores when comparing means with and without these participants included.’ However, I did wonder whether using means actually replicates the problem to which this paper objects. Therefore, would it be prudent to subject the analysis of this subgroup of movers to the type of analysis the paper later demands?

There is effective use of data from study 1 to illustrate the ‘problem’ of masked effects when relying on means. Equally, the interim discussion promotes a necessary focus on this issue and ways around it, however I do wonder whether presenting something a little more tangible just here – in addition to this discussion – might help convince some ‘more traditional’ readers, e.g. providing percentages of participants whose test scores improved by more than a specified amount (or at least signposting these as they appear later on).

The following sections, beginning with the new hypotheses, appear to provide a convincing case for the data and analyses presented and are suitably clear and carefully described.

The Discussion makes a clear argument for adopting the statistical approach described here and also appears to point to some flexibility over the choice of variables to be included in calculating Meta Impact: ‘researchers could use the Meta-Impact analysis to gather data on several ‘possible’ variables’. Is it possible to provide some indications of alternative variables, particularly where the availability of psychometric testing may be limited, e.g. specifying workplace behaviours, etc?

Overall, the enthusiasm for a new approach is expressed effectively and hopefully will indeed inspire others to take the much-needed steps shown by the authors. Thank you for showing the way!

On a minor point, there were some typos which I’ve listed below:

Table 1 – ‘backward’ or ‘backwards’ digit span?

On p24, should it be ‘1-5 and 0-3’ rather than ‘3-0’?

Please note typo: ‘…WM training has be widely criticised…’ (p25)

and ‘…using in a Likert scales with four or five intervals (p26)

Table 7 – ‘121’ needs making clear that this refers to condition

Graph 2 – perhaps it could be made clearer what the asterisk represents?

Table 9 - Good to have included two examples with details relevant their profiles and trajectories through the study – perhaps a little more comment on ‘idiosyncratic’ is warranted, where Table 9 is referred to at the end of the Results (p.29)?

Reviewer #2: General Comments:

1. I appreciate the invitation to review this interesting manuscript. The Authors address a difficult area in applied research of measuring intervention impact when a study population is small and does not allow matched randomized comparison of intervention and control groups.

2. As noted in the manuscript, the focus on individual experiences that may not be within a general measure of response of a target population may limit the usefulness of the intervention for application in group settings.

3. The research objective is vague as stated in the two paragraphs prior to the Study Design subsection of the manuscript. A concise and focused statement of the study objective would be helpful in understanding the study objective and importance.

4. While the intent of the research is to individualize the effect, effectiveness and possible future value of an intervention, it appears there is still a tendency to subgroup analysis and group-wise comparisons as opposed to true individualized assessment of effect. It would appear that following calculation of score determinations for individuals as described in the “4. Meta-Impact Score” subsection that each individual should be treated as an autonomous measure as opposed to being re-grouped for analysis. The individual analysis component of the research should be more specifically identified and explained.

5. As noted in the discussion of study limitations, the dichotomization fo data is highly suspect for unintentional bias and error.

Specific Comments:

Page 2 or main manuscript: Please delete the phrase “and so on”.

2. “Introduction to the Meta-Impact Score”: The definition of this scoring system as described in this section refers to two main demarcations. Please list these demarcations as the comparison of subgroup improved performance in the intervention and control group is well described, but a second demarcation is not well identified.

6. PLOS authors have the option to publish the peer review history of their article (what does this mean?). If published, this will include your full peer review and any attached files.

Reviewer #1: No

Reviewer #2: **Yes: **Samuel J. Stratton, MD, MPH

---

## [Author Response · Author response to Decision Letter 0]

3 Feb 2022

To the reviewers,

Many thanks for your detailed review and analysis of our work. We recognize that it was not a straightforward paper, yet you have diligently explored the logic of our proposal and you have provided detailed, comprehensive feedback. This will no doubt enhance the paper. Thank you very much for your time and efforts. Below, we address each of your points individually, and the respective changes are marked up in the Main Document, with edits to supporting documents also completed, as required. 

This feedback as available as a separate file also, which may be easier to follow as we have laid out the comments and responses in a table.

Highest regards,

The Authors.

Reviewers' comments:

Reviewer #1

Reviewer #1: Thank you for giving me the opportunity to review this paper. It offers a well-constructed and supported rationale for taking this innovative approach to the analysis and interpretation of data in real-world studies. As such, it is certainly needed and this need is underlined by the challenges facing all those seeking to understand ‘what works?’ It is good to see an individual differences type approach to group-level intervention, however I wonder whether there is also an opportunity to use the examples of advances in other fields – such as medicine – as leverage in the argument that work-based interventions should also move towards a more individually tailored approach. Thank you for your comments, we agree! We’ve added the link to personalized medicine to the stated research intention and a note in the conclusion also. This is a really helpful link and connection to wider research. Have also added the work personalized to the abstract, this phrase ought to attract wider engagement and peer comment.

The data presented clearly exemplify the methodological challenges faced in conducting real-world research, as well as the dilemmas in reaching decisions about whether an intervention works/doesn’t work with a client group. The authors are to be congratulated on the efforts taken in gathering and refining these data sets. Thank you again! Realist, quasi-experimental research is very needed in our opinion, and traditional methods make it hard to get good work published.

A main thrust of the paper appears to be that some participants performed better in some areas than the control group, but there is a claim on page 12 for ‘consistent person-specific indications of improvements in confidence and work-related performance’ in both of the studies described. I suspect it would help support such a statement with clear reference to where this is evidenced and/or what this means in the context of the study, i.e. were these reflected in test results, participant feedback, etc? This could be important as this is one piece of important evidence on which the premise of the rest of the paper is based. Thank you for noting this oversight. We have clarified as the SE measure is not consistent, but the behavioural measures are, they are now duly signposted.

Due care appears to have been taken in describing the various calculations and so it is the Meta-Impact score which becomes the focus of attention. Described on page 13 as ‘an aggregation of predefined levels of individual improvement across a number of different measures of a range of plausible intervention effects’, it is at least partly based on ‘assessing the numerical point at which a ‘Real World Value’ is most likely to have occurred’. This does pose an interesting question about what this ‘numerical point’ might be and I wonder whether this paper can afford to be bolder in suggesting it holds the key to how this figure might be derived, using the statistical processes described later? Alternatively, I wonder if there is also a risk that the thorny issue of when a difference on paper equates to a difference in behaviour may get in the way of what the paper is hoping to convey. Perhaps signposting other papers who do so, might help at this point? Agreed – clarified and signposted to an excellent debate that is emerging due to new media technologies on the value of data as a proxy for behaviour.

In terms of the study design, it seems reasonable to have made ‘refinements to measures and protocol to ascertain if the mixed results were due to artefacts or if the results would be replicated in a new sample’, however it does raise the question of what these might be. Clarified.

It is not much of a loose end in itself, but in the later context of moving clients from one group to another, these may be seen as potential ‘liberties’ in the eyes of some readers. Care is taken to state there were ‘no significant differences in pre or post group mean scores when comparing means with and without these participants included.’ However, I did wonder whether using means actually replicates the problem to which this paper objects. Therefore, would it be prudent to subject the analysis of this subgroup of movers to the type of analysis the paper later demands? Well this is interesting because you are right, and they do contribute a Meta Impact score. One can argue here that they moved because they had urgent needs and that their self-selection into a group where they could be immediately helped to improve reflected their self-awareness. I have acknowledged the issue in the text and noted its commonality in human sciences.

There is effective use of data from study 1 to illustrate the ‘problem’ of masked effects when relying on means. Equally, the interim discussion promotes a necessary focus on this issue and ways around it, however I do wonder whether presenting something a little more tangible just here – in addition to this discussion – might help convince some ‘more traditional’ readers, e.g. providing percentages of participants whose test scores improved by more than a specified amount (or at least signposting these as they appear later on). Agreed – have signposted and summarized.

The following sections, beginning with the new hypotheses, appear to provide a convincing case for the data and analyses presented and are suitably clear and carefully described.

The Discussion makes a clear argument for adopting the statistical approach described here and also appears to point to some flexibility over the choice of variables to be included in calculating Meta Impact: ‘researchers could use the Meta-Impact analysis to gather data on several ‘possible’ variables’. Is it possible to provide some indications of alternative variables, particularly where the availability of psychometric testing may be limited, e.g. specifying workplace behaviours, etc? Agreed, have made suggestions.

Overall, the enthusiasm for a new approach is expressed effectively and hopefully will indeed inspire others to take the much-needed steps shown by the authors. Thank you for showing the way! Thank you again!

On a minor point, there were some typos which I’ve listed below:

Table 1 – ‘backward’ or ‘backwards’ digit span? – as stated, backward is correct

On p24, should it be ‘1-5 and 0-3’ rather than ‘3-0’? Corrected

Please note typo: ‘…WM training has be widely criticised…’ (p25) Corrected 

and ‘…using in a Likert scales with four or five intervals (p26) Corrected 

Table 7 – ‘121’ needs making clear that this refers to condition Corrected 

Graph 2 – perhaps it could be made clearer what the asterisk represents? Corrected

Table 9 - Good to have included two examples with details relevant their profiles and trajectories through the study – perhaps a little more comment on ‘idiosyncratic’ is warranted, where Table 9 is referred to at the end of the Results (p.29)? Clarified

Reviewer #2: 

1. I appreciate the invitation to review this interesting manuscript. The Authors address a difficult area in applied research of measuring intervention impact when a study population is small and does not allow matched randomized comparison of intervention and control groups. Thank you for these comments. We agree that there are a lot of overlapping priorities in evaluative, realist research. 

2. As noted in the manuscript, the focus on individual experiences that may not be within a general measure of response of a target population may limit the usefulness of the intervention for application in group settings We hope that our proposed method can be refined and lead to a wider discussion on the benefits of individual and group level analysis in a range of settings. As personalized treatment pathways start to become more popular in medicine, clinical work and applied field generally, we hope this gives researcher some ideas on balancing the two when conducting research that will drive policy. Have added this clarity to the conclusion.

3. The research objective is vague as stated in the two paragraphs prior to the Study Design subsection of the manuscript. A concise and focused statement of the study objective would be helpful in understanding the study objective and importance. Agreed, statement added, thank you.

4. While the intent of the research is to individualize the effect, effectiveness and possible future value of an intervention, it appears there is still a tendency to subgroup analysis and group-wise comparisons as opposed to true individualized assessment of effect. It would appear that following calculation of score determinations for individuals as described in the “4. Meta-Impact Score” subsection that each individual should be treated as an autonomous measure as opposed to being re-grouped for analysis. The individual analysis component of the research should be more specifically identified and explained. Agreed – text added to draw this out and make it more clear that there are two parts to stage 4.

5. As noted in the discussion of study limitations, the dichotomization fo data is highly suspect for unintentional bias and error.

 Minor edits on this section, it is broader than the remit of this article but indeed deserving of clear acknowledgement.

Page 2 or main manuscript: Please delete the phrase “and so on”. Corrected

“Introduction to the Meta-Impact Score”: The definition of this scoring system as described in this section refers to two main demarcations. Please list these demarcations as the comparison of subgroup improved performance in the intervention and control group is well described, but a second demarcation is not well identified. 

 Corrected

---

## [Editor Report · Decision Letter 1]

1 Mar 2022

Does it work? Using a Meta-Impact score to examine global effects in quasi-experimental intervention studies.

PONE-D-21-08221R1

Dear Dr. Doyle,

We’re pleased to inform you that your manuscript has been judged scientifically suitable for publication and will be formally accepted for publication once it meets all outstanding technical requirements.

Kind regards,

Guest Editor

PLOS ONE

Additional Editor Comments (optional):

Dear Author

Thank you so much for making the necessary amendments to your paper in accordance with the suggestions of both reviewers. I have noted a small number of minor errors (please see below) which you might like to address in the final version for publication. I have relayed my decision as guest editor to the senior editor with whom the final decision to publish rests.

all best wishes

Guest editor

Page 13, paragraph 1: the two sentences at the end of this paragraph should be amalgamated, so it reads, ‘…The low correlations…provided…etc’.

Page 19, paragraph 2: please check it and it’s are both intended in sentence which reads, ‘…tangible effect it creates its ‘Real World Value’.

Page 20, in section headed, ‘Defending a dichotomization’, please amend to indicates in sentence which reads, ‘Note that at the individual level of analysis, the presence of a score of =/>1, indicate success…’

Page 22, end of first paragraph: please delete ‘and’ from last sentence which currently reads, ‘Self-efficacy did not seem to contribute much to the Meta-Impact (only one improvement) with and all other measures improving…’

Page 23, in section headed, ‘CS2 Results’, please separate words as currently reads, ‘S8 Tableshows the…’

Page 23, line 6 of same section contains, ‘eight (out of 26; )’ – please amend.

Page 25, where it states, ‘Our improvement score delineation facilitates nuance at the group level by sacrificing it at the individual level’. It might just be me, but please check this is the way round you intend it to read, and whether indeed much has been sacrificed, or rather prioritised?

Page 27, line 5: should it read as, ‘…between intervention and control groups measure’ or ‘measured’?

References

2. Please relocate ‘p’ at end of reference

10. Is the journal title missing?

13. Is ‘BPS editor’ correct? If you prefer, there is a weblink: https://www.bps.org.uk/news-and-policy/psychology-work-improving-wellbeing-and-productivity-workplace

51. Should title be in block capitals?
---

## [Editor Report · Acceptance letter]

7 Mar 2022

PONE-D-21-08221R1 

Does it work? Using a Meta-Impact score to examine global effects in quasi-experimental intervention studies. 

Dear Dr. Doyle:

I'm pleased to inform you that your manuscript has been deemed suitable for publication in PLOS ONE. Congratulations! Your manuscript is now with our production department. 

Kind regards, 

on behalf of

Dr. Ashley Weinberg 

Guest Editor

PLOS ONE